

# Whole-genome sequence and genesis of an avian influenza virus H5N1 isolated from a healthy chicken in a live bird market in Indonesia: accumulation of mammalian adaptation markers in avian hosts

Saifur Rehman[1], Rima Ratnanggana Prasetya[2], Krisnoadi Rahardjo[2], Mustofa Helmi Effendi[1], Fedik Abdul Rantam[3], Jola Rahmahani[3], Adiana Mutamsari Witaningrum[1], Aldise Mareta Nastri[2], Jezzy Renova Dewantari[2], Yasuko Mori[4] and Kazufumi Shimizu[2,4]

[1] Division of Veterinary Public Health, Faculty of Veterinary Medicine, Airlangga University, Surabaya, East Java, Indonesia
[2] Indonesia-Japan Collaborative Research Center, Institute of Tropical Disease, Airlangga University, Surabaya, East Java, Indonesia
[3] Laboratory of Virology and Immunology, Division of Microbiology, Faculty of Veterinary Medicine, Airlangga University, Surabaya, East Java, Indonesia
[4] Center for Infectious Diseases, Graduate School of Medicine, Kobe University, Kobe, Japan

Corresponding authors
Mustofa Helmi Effendi,
mhelmieffendi@gmail.com
Kazufumi Shimizu,
shimizu.kazufumi@gmail.com

## ABSTRACT

**Background:** Influenza A viruses are a major pathogen that causes significant clinical and economic harm to many animals. In Indonesia, the highly pathogenic avian influenza (HPAI) H5N1 virus has been endemic in poultry since 2003 and has caused sporadic deadly infections in humans. The genetic bases that determine host range have not yet been fully elucidated. We analyzed the whole-genome sequence of a recent H5 isolate to reveal the evolution toward its mammalian adaptation.

**Methods:** We determined the whole-genome sequence of A/chicken/East Java/ Av1955/2022 (hereafter, "Av1955") from a healthy chicken in April 2022 and conducted phylogenetic and mutational analysis.

**Results:** Phylogenetic analysis revealed that Av1955 belonged to the H5N1 clade 2.3.2.1c (Eurasian lineage). The six gene segments (PB1, PB2, HA, NP, NA, and NS) out of the eight segments derived from viruses of H5N1 Eurasian lineage, one (PB2) from the H3N6 subtype and the remaining one (M) from the H5N1 clade 2.1.3.2b (Indonesian lineage). The donor of the PB2 segment was a reassortant among three viruses of H5N1 Eurasian and Indonesian lineages and the H3N6 subtype. The HA amino acid sequence contained multiple basic amino acids at the cleavage site. Mutation analysis revealed that Av1955 possessed the maximal number of mammalian adaptation marker mutations.

**Conclusions:** Av1955 was a virus of H5N1 Eurasian lineage. The HA protein contains an HPAI H5N1-type cleavage site sequence, while the virus was isolated from a healthy chicken suggesting its low pathogenicity nature. The virus has increased mammalian adaptation markers by mutation and intra- and inter-subtype reassortment, gathering gene segments possessing the most abundant maker mutations among previously circulating viruses. The increasing mammalian adaptation mutation in avian hosts suggests that they might be adaptive to infection

in mammalian and avian hosts. It highlights the importance of genomic surveillance and adequate control measures for H5N1 infection in live poultry markets.

## INTRODUCTION

Influenza A viruses are a major pathogen that causes significant clinical and economic harm to various species, including poultry, pigs, horses, marine mammals, and humans (*Peiris, de Jong & Guan, 2007*; *Webster et al., 1992*). The surface antigenicity of the virus particles divides them into 18 hemagglutinin (HA) (H1 to H18) and 11 neuraminidase (NA) (N1 to N11) subtypes (*Tong et al., 2013*). Highly pathogenic avian influenza (HPAI) H5N1 viruses are capable of sporadic human infection. They have the potential to cause severe sickness with a high case fatality rate among people who have been hospitalized and proven to have the virus. To date, there have been 861 confirmed disease cases in people, and 455 have led to death (*WHO H5N1, 2021*). Indonesia reported 200 confirmed human cases and 168 fatalities. The 168 fatalities account for over one-third of all deaths among all affected countries. However, Indonesia had only two cases in 2014 and 2015, zero in 2016, and one in 2017, the last (*World Health Organization, 2015*; *World Health Organization, 2019*; *WHO H5N1, 2021*).

The HPAI H5N1 subtype was first discovered in Indonesia in 2003 (*Li et al., 2004*), spreading to numerous regions and killing over 16 million chickens by the end of 2007 (*Lam et al., 2008*; *Putri, Widyarini & Asmara, 2019*). The sequence of the hemagglutinin gene classified it into clade 2.1. This clade subsequently branched into clades 2.1.1 to 2.1.3 in Indonesia (*WHO/OIE/FAO H5N1 Evolution Working Group, 2008*). Clade 2.1.3 further branched into clades 2.1.3.1 to 2.1.3.3; then clade 2.1.3.2 into clades 2.1.3.2a and b (*WHO/OIE/FAO H5N1 Evolution Working Group, 2014*). Clade 2.1.1 viruses were predominantly isolated from HPAI-infected chickens during outbreaks between 2003 and 2005. Clades 2.1.2 and 2.1.3 were isolated from birds between 2003 and 2005 and humans in 2005. Clade 2.1.3.2, a branch of clade 2.1.3, became prevalent in poultry and humans in 2007. Clade 2.1.3.2b, a descendant of clade 2.1.3.2, became prevalent from 2010 to 2012.

An incursion of a new HPAI H5N1 clade, clade 2.3.2.1c, from the mainland of Southeast Asia occurred in Indonesia in 2012 (*Dharmayanti et al., 2014*). The clade became prevalent as early as 2013. *Shimizu et al. (2016)* reported the isolations of three avian influenza viruses in East Java, Indonesia: H5N1 clade 2.3.2.1c (Av154) from an HPAI outbreak in a turkey farm in 2013, H5N1 clade 2.1.3.2b (Av240) from an ill chicken at a live poultry market in 2014, and H3N6 (Av39) from a mildly ill duck at a live poultry market in 2013 (*Shimizu et al., 2016*). Clade 2.3.2.1c was designated as Eurasian lineage, while clade 2.1.3.2b was designated as Indonesian lineage based on the place of emergence. Since 2016, only Eurasian lineage had been isolated in East Java, Indonesia (K. Rahardjo, 2019, unpublished data).

Clade 2.3.2.1c caused only one fatal infection in humans in Indonesia, the last case in 2017. The genetic bases that determine host range have not yet been fully elucidated. Influenza virus evolves by the mutation of the genes and by the reassortment of the gene segments. *Mertens et al. (2013)* identified 149 phenotypic marker mutations related to host tropism or increased pathogenicity in mammals from previously reported literature. Recently, *Rehman et al. (2022)* reported detections of avian influenza A/H5 viruses in poultry at live bird markets in East Java, Indonesia. In this study, we determined the whole-genome sequence of one of the detected viruses to identify the clade/lineage and some characteristic features of the amino acid sequences of the viral proteins. We compared the phenotypic marker mutations with its ancestral viruses to reveal the evolution toward its mammalian adaptation.

## MATERIALS AND METHODS

### Ethical approval

The Animal Care and Use Committee at the Faculty of Veterinary Medicine, Universitas Airlangga, Surabaya, Indonesia gave their approval for every step of this study (approval no. 1.KE.028.03.2021).

### Virus

*Rehman et al. (2022)* conducted a cross-sectional study from March 2021 to April 2022, collecting 600 tracheal and cloaca swab samples from live bird markets in East Java, Indonesia. The authors isolated an avian influenza A/H5 virus from a tracheal swab sample collected on 5 March 2022 from a healthy chicken, which showed no symptoms of lethargy, shortened neck, retracted feather, diarrhea, torticollis, and dyspnea. For this research, they provided a one-mL aliquot of an allantoic fluid harvested from the A/H5 virus-infected egg. The harvest was positive for the hemagglutination test with chicken red blood cells. It was also positive for one-step TaqMan real-time RT-PCR tests targeting the influenza A virus M gene and H5 HA gene using sets of primers and probes as previously described (*Shimizu et al., 2016*) (data, not shown). All procedures were performed in the BSL3 laboratory of the Institute of Tropical Disease, Airlangga University. We named the virus A/chicken/East Java/Av1955/2022 (hereafter, "Av1955").

### Whole-genome sequencing

We performed genome analysis by next-generation sequencing as previously described (*Novianti et al., 2019*). Total RNA was isolated from the allantoic harvest using a QIAamp viral Minikit (Qiagen, Hilden, Germany). Linear polyacrylamide was used as a transporter rather than tRNA. A TruSeq RNA sample preparation kit version 2 was utilized to compile an RNA library (Illumina, San Diego, CA, USA). The library was loaded in the flow cell of the 300-cycle MiSeq reagent kit version 2 (Illumina, San Diego, CA, USA). A MiSeq system (Illumina, San Diego, CA, USA) sequenced the barcode-labeled multiplex library, which had two runs of 150 bp each. The system created the FASTQ files of sequence reads removing the primer and adaptor sequences. For analysis, the files were imported into CLC Genomics Workbench version 8.1 (CLC bio, Osaka, Japan). The reads were mapped to the
genomes of 27 reference viruses of influenza type A virus, including all subtypes of HA (H1 to H18) and NA (N1 to N11). Tentative complete eight-segment genome sequences were constructed from the assembled consensus with the best coverage and common sequences of type A influenza viruses at the 5′ end (12 nucleotides (nt)) and 3′ end (13 nt) of the genome segments. The reads were mapped again to the tentative complete genome sequence to assemble Av1955 consensus sequences. The assembled sequences covered 97.4% to 99.5% of the complete genome-segments mostly lacking the 5′ and 3′ consensus sequences. The sequences of the eight genome-segments were submitted to the GISAID database with isolate ID: EPI_ISL_13690275.

### Genetic analysis

BLAST analysis in the GISAID EpiFlu database (https://gisaid.org) on 10 August 2022 identified viruses with the highest identity to the nucleotide sequence of one or more gene segments of Av1955.

The genetic information processing software Genetyx v14 (Genetyx Co., Tokyo, Japan) generated the phylogenetic trees using the RAxML with 100 bootstrap replicates. The trees are rooted in the A/South Carolina/1/1918(H1N1) (HA: GenBank AF117241) or A/Brevig Mission/1/1918(H1N1) (NA: AF250356, PB2: DQ208309, PB1: DQ208310, PA: DQ208311, NP: AY744935, M: AY130766, NS: AF333238).

The amino acid sequences of the viral proteins were decoded from the genome nucleotide sequences and analyzed for nonsynonymous mutations. The putative host adaptation mutations were analyzed according to the evaluation of phenotypic markers described by Mertens et al. (2013).

### Isolate ID (EPI_ISL) in the GISAID database of viruses used as a reference

A/duck/East Java/Av39/2013 (H3N6) (hereafter, Av39): EPI_ISL_307026; A/chicken/East Java/Av240/2014 (H5N1 Indonesian lineage) (HA clade 2.1.3.2b) (hereafter, Av240): EPI_ISL_307019; A/turkey/East Java/Av154/2013 (H5N1 Eurasian lineage) (HA clade 2.3.2.1c) (hereafter, Av154): EPI_ISL_307002; A/eagle/Jakarta Timur/20616-206-III/2016 (H5N1) (hereafter, Eg20616): EPI_ISL_266799; A/chicken/East Java/Av1210/2017 (H5N1) (hereafter, Av1210): EPI_ISL_401823; A/chicken/East Java/Spg119/2018 (H5N1) (hereafter, Spg119): EPI_ISL_365528; and A/chicken/East Java/Av1534/2019 (H5N1) (hereafter, Av1534): EPI_ISL_401824.

We noted the accession number of each genome segment of the viruses in Table S1.

## RESULTS

### HA and NA subtypes of Av1955

The BLAST analysis revealed that the HA sequence of Av1955 was most close to Spg119 of H5N1 (the sharing identity, 97.7%) and the NA to Av1210 of H5N1 (97.3%) among the viruses in the GISAD database (Table 1). The HA and NA segments were most close to Av154 of H5N1 Eurasian lineage (HA clade 2.3.2.1c) (95.1% and 96.2%, respectively) among the three isolates (Av39, Av240, and Av154) during 2013–2014 in East Java

**Table 1 Viruses with the highest identity to the nucleotide sequence of one or more gene segments of Av1955.**

| Genome segment | Identity to the eight genome segments of Av1955, %[a] | | | | | | | |
| --- | --- | --- | --- | --- | --- | --- | --- | --- |
| | Isolates in 2013–2014[b] | | | Viruses with the highest identity[b] | | | | |
| | Av39 | Av240 | Av154 | Eg20616 | Av1210 | Spg119 | Av1534 | |
| | H3N6 | H5N1-Ind | H5N1-Eur | H5N1 | H5N1 | H5N1 | H5N1 | Av1955 |
| PB2 | **93.2** | 85.2 | 85.3 | 85.2 | 85.2 | **98.0** | 85.0 | 100.0 |
| PB1 | 87.6 | 91.3 | **95.9** | 96.0 | 97.4 | 96.7 | **97.9** | 100.0 |
| PA | 90.9 | 87.6 | **96.0** | 95.3 | 96.8 | 95.4 | **98.2** | 100.0 |
| HA | 53.8 | 87.0 | **95.2** | 94.6 | 94.4 | **97.7** | 93.8 | 100.0 |
| NP | 90.3 | 93.5 | **97.0** | 96.7 | 96.6 | **98.3** | 96.6 | 100.0 |
| NA | 54.0 | 87.7 | **96.2** | 95.0 | **97.3** | 96.0 | 95.5 | 100.0 |
| M | 90.4 | **98.1** | 93.8 | **98.9** | 93.3 | 97.0 | 93.4 | 100.0 |
| NS | 88.8 | 93.0 | **97.2** | 92.4 | 97.9 | **98.5** | 97.2 | 100.0 |

Notes:
[a] "Bold" indicates the highest identity to Av1955 among Av39, Av154, and Av240. Bold and underlined indicate the highest identity to Av1955 among viruses in the EpiFlu database of GISAID with whole-genome sequence.
[b] Virus short name: Av39, A/duck/East Java/Av39/2013; Av240, A/chicken/East Java/Av240/2014; Av154, A/turkey/East Java/Av154/2013; Eg20616, A/eagle/Jakarta Timur/20616-206-III/2016; Av1210, A/chicken/East Java/Av1210/2017; Spg119, A/chicken/East Java/Spg119/2018; and Av1534, A/chicken/East Java/Av1534/2019.

(Table 1). In the phylogenies, the HA and NA segments are on the same branch with Av154 (Fig. 1). These results clearly indicate that Av1955 is a virus of the H5N1 Eurasian lineage (HA clade 2.3.2.1c).

## Genesis of Av1955

The BLAST analysis revealed that the genome sequence of Av1955 was most close to Spg119 for the PB2 segment (the sharing identity, 98.0%), HA (97.7%), NP (98.3%), and NS (98.5%); to Av1534 for the PB1 (97.9%) and PA (98.1%); to Av1210 for the NA (97.3%); and to Eg20616 for the M (98.9%) among the viruses in the GISAD database (Table 1). These results indicated that Av1955 acquired the PB2, HA, NP, and NS segments from an Spg119-like virus; the PB1 and PA from an Av1534-like virus; the NA from an Av1210-like virus; and the M from an Eg2061-like virus. These four putative donors of the segments are H5N1 Eurasian lineage since all HA and NA closely cluster with Av154 of H5N1 Eurasian lineage in the phylogenies of HA and NA (Fig. 1).

Av1955 was most close to Av39 for the PB2 (93.2%); to Av154 for the PB1 (95.9%), PA (96.0%), HA (95.2%), NP (97.0%), NA (96.2%), and NS (97.2%); and Av240 for the M (98.1%) among the three isolates in East Java during 2013–2014. In the phylogenies, the PB2 of Av1955 closely clusters with Spg119 and Av39; all of the PB1, PA, HA, NP, and NA cluster with Eg20616, Av1210, Spg119, Av1534, and Av154; the M with Eg20616, Spg119, and Av154; and the NS with Av1210, Spg119, Av1534, and Av154.

The PB2, M, and NS of Eg20616 closely cluster with Av240, while the other six segments with Av154, indicating that Eg206166 is a reassortant between Av240 and Av154; Eg20616 acquired the PB2, M and NS segments from an Av240-like virus, and the other six segments from an Av154-like virus.

The PB2 and M of Spg119 closely cluster with Av39 and Av240, respectively, while the other six segments with Av154, indicating that Spg119 is a triple reassortant between Av39,

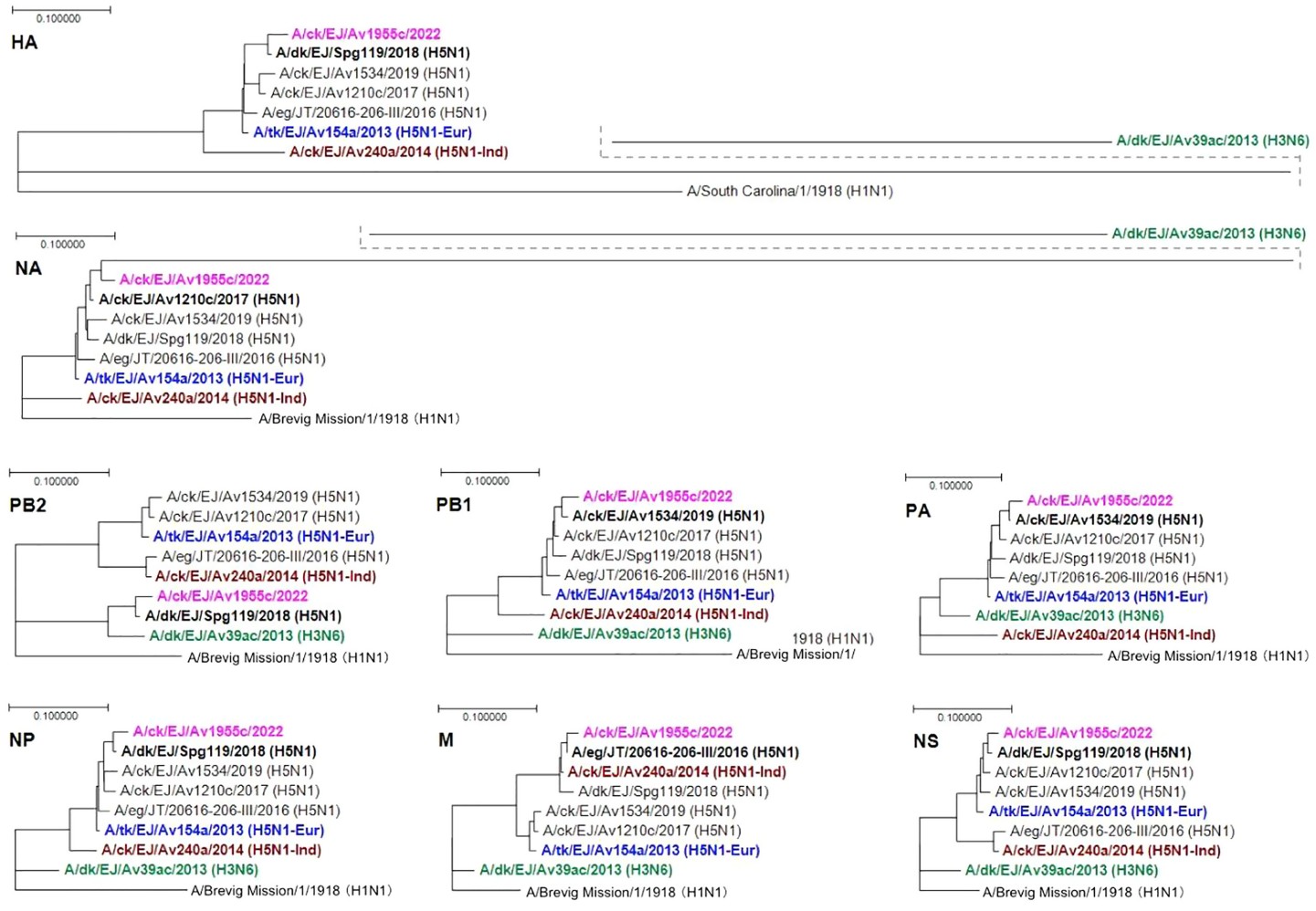

**Figure 1 Phylogenetic analysis of the eight gene segments of Av1955.** The genetic information processing software Genetyx v14 (Genetyx Co., Tokyo, Japan) generated the phylogenetic trees using the RAxML with 100 bootstrap replicates. The tree is rooted in the A/Brevig Mission/1/1918 or A/South Carolina/1/1918 (H1N1). Four viruses with the highest identity to the nucleotide sequence of one or more gene segments of A/ck/EJ/Av1955/2022 (bold and pink color) were included in the phylogenies; these were A/eg/JT/20616-206-III/2016 (H5N1), A/ck/EJ/Av1210c/2017 (H5N1), A/ck/EJ/Spg119/2018 (H5N1) and A/ck/EJ/Av1534/2019 (H5N1). Three viruses isolated from poultry in East Java in 2013–2014 were also included; A/dk/EJ/Av39ac/2013 (H3N6) (bold and green), A/ck/EJ/Av240/2014 (H5N1-Ind) (bold and brown), and A/tk/EJ/Av154/2013 (H5N1-Eur) (bold and blue). Bold and black highlighted the virus at the nearest position to A/ck/EJ/Av1955c/2022 in each of the eight genome segments. "ck": chicken, "eg": eagle, "dk": duck, "tk": turkey, "EJ": East Java, "JT": Jakarta Timur, "–Ind": Indonesian lineage (HA clade 2.1.3.2b), and "-Eur": Eurasian lineage (HA clade 2.3.2.1c).

Av240, and Av154; Spg119 acquired the PB2 segment from an Av39-like virus, the M segment from an Av240-like virus, and the other six segments from an Av154-like virus.

From the described results, we proposed a model of the genesis of Av1955 by multi-steps of inter- and intra-subtype reassortment, as illustrated in Fig. 2.

## Characteristic amino acid sequences of Av1955

Table 2 summarizes the findings of an analysis of Av1955 amino acid sequences of the receptor binding, cleavage, and glycosylation sites in HA, deletions in NA and NS1,

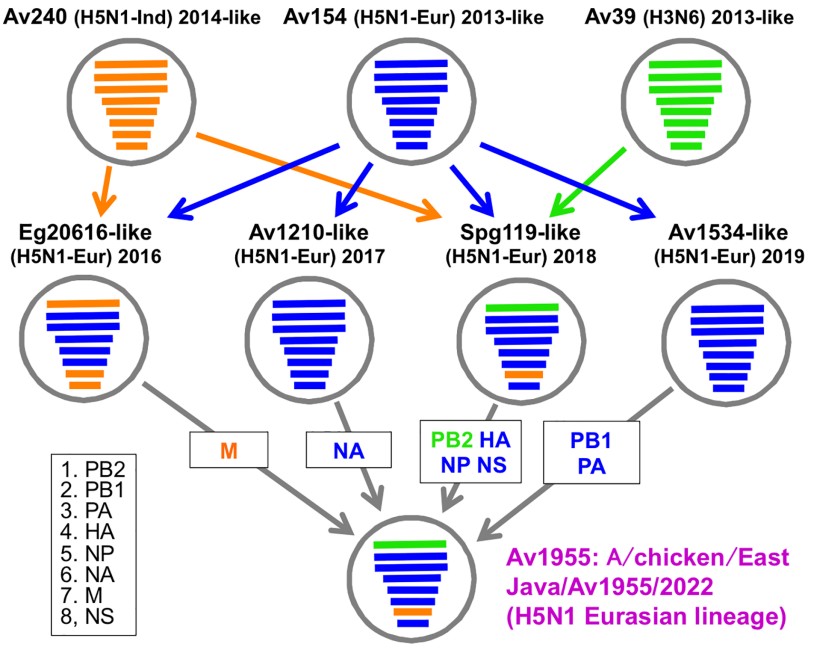

**Figure 2 The genesis of Av1955 by multiple reassortments.** Av39: A/duck/East Java/Av39/2013 (H3N6), Av240: A/chicken/East Java/Av240/2014 H5N1 Indonesian lineage, Av154: A/turkey/East Java/ Av154/2013 (H5N1 Eurasian lineage), Eg20616: A/eagle/Jakarta Timur/20616-206-III/2016 (H5N1 Eurasian lineage), Av1210: A/chicken/East Java/Av1210/2017 (H5N1 Eurasian lineage), Spg119: A/chicken/East Java/Spg119/2018 (H5N1 Eurasian lineage), and Av1534: A/chicken/East Java/Av1534/ 2019 (H5N1 Eurasian lineage). Isolate ID (EPI_ISL) in the GISAID database of the viruses was noted in Materials and Methods.

truncation of PB1-F2, and amantadine and rimantadine resistant mutations in M2. At the receptor binding region of the HA protein, the amino acid sequences were E-186, G-221, Q-222, and G-224 (H5 numbering), which are consistent with the avian type specifically binding 2,3-linked sialic acid receptor. There were no mutations in this region among the seven reference viruses. The amino acid sequence at the cleavage site of the HA protein was PQRE-RRRKR for Av1955, Av154, and the four donor strains and was PQRESRRKKR for Av240; all of these possessed five consecutive basic amino acid residues as typical highly pathogenic avian influenza A viruses. Av39 had PEKQT- - - - R at the site, only one basic amino acid residue R, as specific low pathogenic viruses. A glycosylation site, NST, was present at 154–156 in Av240, while the particular sequence was absent in seven other viruses, including Av1955 as a result of substitution(s) to GST for Av39; DNA for Av154; NNA for Eg20616, Av1210, Spg119, and Av1534; and CNA for Av1955.

There were deletions in NA at 49–68 and NS at 80–84 with Av1955 and all reference viruses except Av39. The Av1955 PB1-F2 was truncated at 25, while that of the reference viruses, except for Av39 and Av240, was truncated at 57. Av39 and Av240 possessed 87 open reading frames for PB1-F2, almost full of 90. The M2 of Av1955 was a resistant type to amantadine and rimantadine.

**Table 2 Characteristic amino acid sequences of Av1955: the receptor binding, cleavage, and glycosylation sites in HA, deletion in NA and NS1, truncation of PB1-F2, and amantadine resistant mutation in M2.**

| Amino acid mutation | | | Amino acid sequence[a] | | | | | | | | Phenotype[b] |
|---|---|---|---|---|---|---|---|---|---|---|---|
| Protein | Mutation | Site, H5 numbering | Isolates in 2013–2014[c] | | | Viruses with the highest identity[c] | | | | | |
| | | | Av39 | Av240 | Av154 | Eg20616 | Av1210 | Spg119 | Av1534 | Av1955 | |
| HA | Receptor binding | E186G/D | E | E | E | E | E | E | E | E | [1] |
| | | G221D | G | G | G | G | G | G | G | G | [2] |
| | | Q222L | Q | Q | Q | Q | Q | Q | Q | Q | [3] |
| | | G224S | G | G | G | G | G | G | G | G | [4] |
| | Multiple basic amino acid | 321-330 Cleavage site | PEKQT- - - - R↓G | PQRES **RRKKR↓**G | PQRE - **RRRKR↓**G | PQRE - **RRRKR↓**G | PQRE - **RRRKR↓**G | PQRE - **RRRKR↓**G | PQRE - **RRRKR↓**G | PQRE - **RRRKR↓**G | [5] |
| | Loss of glycosylation | Loss of 154–156 (NXT/S) | **Lost** (GST) | NST | **Lost** (DNA) | **Lost** (NNA) | **Lost** (NNA) | **Lost** (NNA) | **Lost** (NNA) | **Lost** (CNA) | [6] |
| NA | Deletion | 49–68 | Not deleted | **Deleted** | **Deleted** | **Deleted** | **Deleted** | **Deleted** | **Deleted** | **Deleted** | [7] |
| NS1 | Deletion | 80–84 | TIASV | **Deleted** | **Deleted** | **Deleted** | **Deleted** | **Deleted** | **Deleted** | **Deleted** | [8] |
| PB1-F2 | Truncation | at 12 | at 87 | at 87 | at 57 | at 57 | at 57 | at 57 | at 57 | **at 25** | [9] |
| M2 | Resistance to amantadine | V27A | V | **A** | I | **A** | I | **A** | V | **A** | [10] |
| | | S31/N/G | S | **N** | S | **N** | S | **N** | S | **N** | [11] |

**Notes:**
[a] Bold and underlined indicate the mammalian adaptation markers.
[b] [1], Increased virus binding to α2,6 (*Glaser et al., 2005*); [2], change in receptor binding affinity from avian to human receptors (*Rogers et al., 1983*); [3], change in receptor binding recognition from α2,3 to α2,6 (*Connor et al., 1994*), Airborne transmissible in mammals (*Maines et al., 2011*); [4], increased virus binding to α2,6, (*Imai et al., 2012*), airborne transmissible in mammals (*Maines et al., 2011*); [5], increased virulence in mice (*Klenk & Garten, 1994*; *Suguitan et al., 2012*; *Yudhawati et al., 2020*); [6], increase virus binding to α2,6 and pathogenicity in mice (*Wang et al., 2010*; *Zhang et al., 2015*); [7], enhanced virulence in mice (*Matsuoka et al., 2009*; *Zhou et al., 2009*); [8], enhance virulence in mice associated with D92E shift (*Long et al., 2008*); [9], attenuate virulence in mammals (*Dharmayanti, Ibrahim & Soebandrio, 2010*); [10], reduce susceptibility to amantadine (*Hay et al., 1985*); [11], reduce susceptibility to amantadine and rimantadine (*Belshe et al., 1988*).
[c] Virus short name: See Table 1 note.

## Mammalian adaptation markers in the amino acid sequences of Av1955

*Mertens et al. (2013)* identified 149 phenotypic marker mutations related to the host tropism or increased pathogenicity in mammals from previously reported literature. They distribute to PB2 (34), PB1 (16), PB1-F2 (1), PA (18), HA (38), NP (10), NA (2), M1 (6), M2 (4), NS1 (16), and NS2 (4). We analyzed the presence or absence of these mutations in Av1955 and seven reference viruses. Table 3 summarizes the results showing the substitutions positive for at least one of the eight viruses. There were no marker mutations in NP and NA proteins. Av1955 had 11 markers in PB2, while the donor of Spg119 had 10. The increase indicates that the virus acquired one marker mutation in the PB2 (R318K) by a substitution mutation. A marker increase also occurred with NS1 (seven to eight by P215T).

The markers of the donor segments were the same number as the recipient, Av1955, with PB1 (eight), PA (three), HA (five), M1 (3), M2 (zero), and NS2 (one). The markers in the transferred proteins were the maximum among the four donor viruses except for the HA. The HA donor, Spg119, had five marker mutations in HA, while Av1210 and Av1534

**Table 3 Mammalian adaptation markers (Mertens et al., 2013) in the amino acid sequences of Av1955.**

| Mammalian adapt. marker mutation | | Amino acid at the marker site[b] | | | | | | | | Phenotype[c] |
| --- | --- | --- | --- | --- | --- | --- | --- | --- | --- | --- |
| | | Isolates in 2013–2014[d] | | | Viruses with the highest identity[d] | | | | | |
| Protein | Mutation[a] | Av39 | Av240 | Av154 | Eg20616 | Av1210 | Spg119 | Av1534 | Av1955 | |
| PB2 | T63I | I | I | I | I | I | I | I | I | [1] |
| | L89V | V | V | V | V | V | V | V | V | [1], [2] |
| | G309D | D | D | D | D | D | D | D | D | [1], [2] |
| | R318K | R | R | R | R | R | R | R | K | [2] |
| | T339K | K | T | T | A | T | K | T | K | [1], [2] |
| | Q368R | R | R | Q | R | Q | R | Q | R | [1], [2] |
| | H447Q | Q | Q | Q | Q | Q | Q | Q | Q | [1], [2] |
| | R477G | G | G | G | G | G | G | G | G | [1], [2] |
| | I495V | V | V | V | V | V | V | V | V | [1], [2] |
| | K526R | K | R | K | R | K | K | K | K | [2], [3] |
| | V667I | I | V | V | V | V | V | V | V | [4] |
| | A676T | T | T | T | T | T | T | T | T | [2], [3] |
| | K702R | R | K | K | K | K | R | K | R | [4] |
| | Subtotal | 11 | 9 | 6 | 10 | 7 | 10 | 6 | 11 | |
| PB1 | A3V | V | V | V | V | V | V | V | V | [1], [2] |
| | L13P | P | P | P | P | P | P | P | P | [1], [2], [5] |
| | R207K | K | R | K | K | K | K | K | K | [2] |
| | K328N | N | N | N | N | N | N | N | N | [2] |
| | S375N | N | N | N | N | N | S | N | N | [1], [2], [5] |
| | H436Y | Y | Y | Y | Y | Y | Y | Y | Y | [1], [2] |
| | L473V | V | V | V | V | V | V | V | V | [2] |
| | M677T | T | T | T | T | T | T | T | T | [1] |
| | Subtotal | 8 | 7 | 8 | 8 | 8 | 7 | 8 | 8 | |
| PA | H266R | R | R | R | R | R | R | R | R | [2], [3] |
| | A404S | A | A | S | S | S | S | S | S | [5] |
| | S/A515T | T | T | T | T | T | T | T | T | [2], [3] |
| | Subtotal | 2 | 2 | 3 | 3 | 3 | 3 | 3 | 3 | |
| HA | D94N | D | S | N | N | N | N | N | N | [6], [7] |
| | S133A | G | S | A | A | A | A | A | A | [6] |
| | Q/H/I138L/N | G | L | Q | Q | Q | Q | Q | Q | [1] |
| | I151T | T | I | I | I | I | I | I | I | [6] |
| | N154D | G | N | D | N | N | S | N | G | [4] |
| | S155N | S | S | N | N | N | N | N | N | [2], [6] |
| | T156A | T | T | A | A | A | A | A | A | [4], [6] |
| | T188I | T | T | T | T | I | T | I | T | [6] |
| | K189S/R | N | M | R | K | R | K | R | K | [2], [6] |
| | V210I | I | V | V | V | V | V | V | V | [6] |
| | A263T | S | A | T | T | T | T | T | T | [1] |
| | Subtotal | 2 | 1 | 7 | 5 | 6 | 5 | 7 | 5 | |

(Continued)

| Mammalian adapt. marker mutation | | Amino acid at the marker site[b] | | | | | | | | Phenotype[c] |
|---|---|---|---|---|---|---|---|---|---|---|
| | | Isolates in 2013–2014[d] | | | Viruses with the highest identity[d] | | | | | |
| Protein | Mutation[a] | Av39 | Av240 | Av154 | Eg20616 | Av1210 | Spg119 | Av1534 | Av1955 | |
| M1 | V15I/T | V | I | I | I | I | I | I | I | [1] |
| | N30D | D | D | D | D | D | D | D | D | [1] |
| | T215A | A | A | A | A | A | A | A | A | [1] |
| | Subtotal | 2 | 2 | 3 | 3 | 3 | 3 | 3 | 3 | |
| M2 | L55F | F | L | L | L | L | L | L | L | [4] |
| | Subtotal | 1 | 0 | 0 | 0 | 0 | 0 | 0 | 0 | |
| NS1 | A/P42S | S | S | S | S | S | S | S | S | [1], [8] |
| | D87E | D | D | E | D | E | E | E | E | [1] |
| | T/D92E | E | E | E | E | E | E | E | E | [1] |
| | L98F | F | F | F | F | F | F | F | F | [1] |
| | I101M | M | M | M | M | M | M | M | M | [1] |
| | T127N | N | A | T | A | T | T | T | T | [1] |
| | V149A | A | A | A | A | A | A | A | A | [1], [8] |
| | N200S | S | G | D | G | S | S | G | S | [9] |
| | P215T | P | P | P | P | P | P | P | T | [5] |
| | Subtotal | 7 | 5 | 6 | 5 | 7 | 7 | 6 | 8 | |
| NS2 | T48A | A | A | T | A | A | A | A | A | [8] |
| | Subtotal | 1 | 1 | 0 | 1 | 1 | 1 | 1 | 1 | |
| Total | | 34 | 27 | 33 | 35 | 35 | 36 | 34 | 39 | |

**Notes:**
[a] H5 numbering.
[b] Bold and underlined indicate the mammalian adaptation markers. Bold, underlined, and italic indicate the mammalian adaptation markers on the genome segment donor.
[c] [1], Pathogenic in mice, increased virulence in mammals; [2], increased polymerase activity, increased replication in mammals; [3], increased virulence in mammals and birds; [4], enhanced transmission, airborne transmissibility in mammals; [5], human host marker; mammalian host marker; [6], increased virus binding to α2,6; [7], enhanced virus fusion; [8], antagonism of IFN induction, escape of antiviral host response; [9], decreased IFN antagonism.
[d] Virus short name: See Table 1 note.

had six and seven, respectively. Av1955 acquired 39 mammalian adaptation markers in its proteins, an 8% increase from 36 of Spg119 (the donor of three genome segments).

## DISCUSSION

Sequencing and phylogenetic analysis (Fig. 1) identified the HA and NA of Av1955 (A/chicken/East Java/Av1955/2022) as an H5N1 subtype of a Eurasian lineage (HA clade 2.3.2.1c). However, Av1955 possesses the PB2 and M genome segments derived from viruses of the H3N6 subtype and H5N1 Indonesian lineage, respectively. The phylogenies indicated that the donor of the PB2 segment, Spg119, was a triple-reassortant among H3N6 and H5N1 Indonesian and Eurasian lineages. The donor of the M segment, Eg20616, was a double-reassortant between H5N1 Indonesian and Eurasian lineages. The genesis of Av1955 might involve two steps of genome-segment reassortment (Fig. 2). It is worth noting that the M genome segment of the Indonesian lineage has remained in

the virus of Eurasian lineage after the 7-year disappearance of the H5N1 Indonesian lineage; the last isolate was in 2015 in East Java (A. Nastri, 2019, unpublished data).

There were no mammalian adaptation mutations in the receptor binding region of the HA protein among the seven reference viruses and Av1955. Their amino acid sequences were E-186, G-221, Q-222, and G-224 (H5 numbering), consistent with the avian type specifically binding to a 2,3-linked sialic acid receptor on the cell surface of avian hosts. If the mutations occur, viruses are unable to infect avian hosts. The HA protein of Av1955 contains an HPAI H5N1-type cleavage site sequence with multiple basic amino acid residues (PQRERRRKR). In contrast, the virus was isolated from a healthy chicken suggesting its low pathogenicity nature. All of the H5N1 isolates in East Java, Indonesia, since 2013 were from sick poultry except a few reassortants possessing a PB2 genome segment derived from avirulent subtypes like H3N6; Spg119, A/chicken/East Java/Spg119/2018 (H5N1), was one of the exceptions isolated from a healthy chicken (K. Rahardjo, 2019, unpublished data). The virus obtained its PB2 genome from an H3N6 virus, as shown in Fig. 2. Av1955 lost a glycosylation site at 154–156 on the HA protein; the loss has been reported to increase pathogenicity in mice (*Suguitan et al., 2012*). Av1955 has deletions at 49–68 on the NA and 80–84 on the NS1, both of which have been shown to increase virulence in mice (*Matsuoka et al., 2009*; *Zhou et al., 2009*; *Long et al., 2008*). The Av1955 PB1-F2 truncated at 25 of the total length, 90. The deletion of the main body has been related to increased virulence in mammals (*Mettier et al., 2021*). Av1955 has amino acid residues of A-27 and N-31 of the M2 protein, which are known to render the phenotype of resistance to amantadine and rimantadine (*Belshe et al., 1988*; *Lan et al., 2010*; *Pinto, Holsinger & Lamb, 1992*). The M genome segment originated from the Av240-like virus of the H5N1 Indonesian lineage (HA clade 2.1.3.2b). Av240 had already acquired the resistant mutations, V27A and N31N, while one of the ancestors of Indonesian lineage, A/Indonesia/5/2005 (H5N1) (HA clade 2.1.3.2), had not yet. It was reported that these mutations occurred during 2003–2008 in human and avian infections in Indonesia (*Dharmayanti, Ibrahim & Soebandrio, 2010*). Since amantadine had been subscribed to influenza patients but never used for poultry in Indonesia, the avian viruses possessing the M segment with resistant mutations might come from human-to-avian transmission.

Av1955 possesses 39 mammalian adaptation marker mutations out of 149 identified by *Mertens et al. (2013)*, while the four viruses of the genome-segment donor have 34–36 and the three ancestral viruses isolated during 2013–2014 have 27–34. The virus has increased the number of the markers by substitution mutations and intra- and inter-subtype reassortment gathering gene segments possessing the most abundant maker mutations among circulating viruses. The increasing mammalian adaptation mutation in avian hosts suggests that they might be adaptive to infection in mammalian and avian hosts.
It highlights the importance of genomic surveillance and adequate control measures for H5N1 infection in live poultry markets. In contrast, we have not identified the remaining 110 mammalian adaptation marker mutations in the eight avian viruses analyzed in this study. One of the most studied mutations, PB2 E627K, has been known to release the restriction of PB2 activities in replication in mammalian cells (*Subbarao, London &*

*Murphy, 1993*), including human volunteers (*Clements et al., 1992*). It is a plausible explanation that this kind of mammalian adaptation mutation arises only in the mammalian host after the transmission.

## CONCLUSIONS

Av1955 is a virus of the H5N1 Eurasian lineage (HA clade 2.3.2.1c). The virus has increased mammalian adaptation markers by substitution mutation and intra- and inter-subtype reassortment, gathering gene segments possessing the most abundant maker mutations among circulating viruses. The increasing mammalian adaptation mutation in avian hosts suggests that they might be adaptive to infection in mammalian and avian hosts. It highlights the importance of genomic surveillance and adequate control measures for H5N1 infection in live poultry markets.

## ACKNOWLEDGEMENTS

We gratefully acknowledge the authors and the originating and submitting laboratories of the sequences from GISAID's EpiFlu database (Table S1), on which this research is based. We are thankful for the assistance of Virology and Biomolecular Laboratory staff, Faculty of Veterinary Medicine Universitas Airlangga were acknowledged for their cooperation in the study. We sincerely thank the Ministry of Research, Technology and Higher Education of Indonesia for their advice to carry out this study in Indonesia.

### Funding

This work and APC were fully funded by the Penelitian Hibah Mandat funding from Universitas Airlangga, Indonesia in the fiscal year 2022, with grant number: 220/UN3.15/PT/2022 and was partly supported by the Japan Agency for Medical Research and Development (AMED), the Japan Program for Infectious Diseases Research and Infrastructure (Global Research Infrastructure) under grant number JP 21wm0125009. The funders had no role in study design, data collection and analysis, decision to publish, or preparation of the manuscript.

### Grant Disclosures

The following grant information was disclosed by the authors:
Penelitian Hibah Mandat funding from Universitas Airlangga, Indonesia: 220/UN3.15/PT/2022.
Japan Agency for Medical Research and Development (AMED).
Japan Program for Infectious Diseases Research and Infrastructure (Global Research Infrastructure): JP 21wm0125009.

### Competing Interests

The authors declare that they have no competing interests.

## Author Contributions

- Saifur Rehman conceived and designed the experiments, performed the experiments, analyzed the data, prepared figures and/or tables, authored or reviewed drafts of the article, and approved the final draft.
- Rima Ratnanggana Prasetya analyzed the data, prepared figures and/or tables, authored or reviewed drafts of the article, and approved the final draft.
- Krisnoadi Rahardjo analyzed the data, prepared figures and/or tables, authored or reviewed drafts of the article, and approved the final draft.
- Mustofa Helmi Effendi conceived and designed the experiments, performed the experiments, authored or reviewed drafts of the article, acquired funding, and approved the final draft.
- Fedik Abdul Rantam conceived and designed the experiments, authored or reviewed drafts of the article, and approved the final draft.
- Jola Rahmahani conceived and designed the experiments, performed the experiments, analyzed the data, authored or reviewed drafts of the article, and approved the final draft.
- Adiana Mutamsari Witaningrum conceived and designed the experiments, authored or reviewed drafts of the article, and approved the final draft.
- Aldise Mareta Nastri analyzed the data, authored or reviewed drafts of the article, and approved the final draft.
- Jezzy Renova Dewantari analyzed the data, authored or reviewed drafts of the article, and approved the final draft.
- Yasuko Mori conceived and designed the experiments, analyzed the data, authored or reviewed drafts of the article, acquired funding, and approved the final draft.
- Kazufumi Shimizu conceived and designed the experiments, performed the experiments, analyzed the data, prepared figures and/or tables, authored or reviewed drafts of the article, and approved the final draft.

## Ethics

The following information was supplied relating to ethical approvals (*i.e.*, approving body and any reference numbers):

The Animal Care and Use Committee at the Faculty of Veterinary Medicine, Universitas Airlangga, Surabaya, Indonesia gave their approval for every step of this study (Approval no. 1.KE.028.03.2021).

## DNA Deposition

The following information was supplied regarding the deposition of DNA sequences:

Isolate ID (EPI_ISL) in the GISAID database of viruses used as a reference:

- A/duck/East Java/Av39/2013 (H3N6) (hereafter, Av39): EPI_ISL_307026,
- A/chicken/East Java/Av240/2014 (H5N1 Indonesian lineage) (HA clade 2.1.3.2b) (hereafter, Av240): EPI_ISL_307019,
- A/turkey/East Java/Av154/2013 (H5N1 Eurasian lineage) (HA clade 2.3.2.1c) (hereafter, Av154): EPI_ISL_307002,

- A/eagle/Jakarta Timur/20616-206-III/2016 (H5N1) (hereafter, Eg2061): EPI_ISL_266799,
  - A/chicken/East Java/Av1210/2017 (H5N1) (hereafter, Av1210): EPI_ISL_401823,
  - A/chicken/East Java/Spg119/2018 (H5N1) (hereafter, Spg119): EPI_ISL_365528,
  - A/chicken/East Java/Av1534/2019 (H5N1) (hereafter, Av1534): EPI_ISL_401824.

## Data Availability

The raw measurements are available in the Supplemental Files.

## Supplemental Information

Supplemental information for this article can be found online at http://dx.doi.org/10.7717/peerj.14917#supplemental-information.

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
