# Peer review of "Whole-genome sequence and genesis of an avian influenza virus H5N1 isolated from a healthy chicken in a live bird market in Indonesia: accumulation of mammalian adaptation markers in avian hosts"

_PeerJ, doi:10.7717/peerj.14917_

## Round 0.1 · original submission · Major Revisions

the present manuscript needs some work as highlighted by reviewer 1. Please provide relevant references, lineages, and sequences in a revised version.

Reviewer 1 ·

Basic reporting

It very important topic of Identifictaion of HPAI H5N1 in healthy Chiken.Author reported clade 2.3.2.1c (Eurasian lineage). The appropriate referances are quoted however need referances for following statements .
Line no 80 Indonesia had only two cases in 2014 and 2015,
81 zero in 2016, and one in 2017,
What about Outbreaks in Poultry and what clade was in circulation through out these yers
clade 2.1.3.2b (Indonesian lineage)....
line no 97 East Java, Indonesia: H5N1 clade 2.3.2.1c (Av154) from an HPAI outbreak in a turkey farm in
98 2013, H5N1 clade 2.1.3.2b (Av240) from an ill chicken at a live poultry market in 2014, and
99 H3N6 (Av39) from a mildly ill duck at a live poultry market in 2013

Experimental design

Method
Virus Pl calrify the samples were collected under any surveillance ? In what frequency ?
Genetic analysis
the basis of the root of the tree for Ha and other gene . It is a H5N1 Gs/Guangdong/1/96 will be appropriate
Blast results What is Spg119 pl provide detailed information .
Av154 of H5N1 Eurasian lineage (HA clade 2.3.2.1c) 2013 outbreak virus in Turky ? What are other closest strains from Global ?
Phylogenetic tree : At leat full treee with representation from other clade for HA and NA can be given and for other segment can be supplimentry figure
Gensis of AV1955 : The strain names are so confusing are they all from Indonecia .use of influenza nomenclature will be useful
Characteristic amino acid sequences of Av1955
acid sequence at the cleavage site of the HA protein was PQRE-RRRKR. Not sure HPAI virus has healthy clinical picture .Kindly support the stetment with proper justification and regerances

Validity of the findings

Conclusion are well stated however rsults need to be discussed more thorouly specialy the human adapoptation marker,sevarity markers

Reviewer 2 ·

Basic reporting

In this manuscript, the authors presented their study about the increase in mammalian adaptation mutation in avian hosts, highlighting the importance of genomic surveillance and adequate control measures for avian influenza viruses in healthy live poultry markets. This research article gives valuable information on the phenotypic marker mutation to host tropism. Also, the authors performed a detailed analysis of the mutations in all eight segments by whole genome sequencing, which may be associated with a needed adaptability for sustenance locally for efficient human transmissibility. The technical methods of the study are straightforward and clear to support the findings.

Experimental design

Minor Comments:
The authors have described well the methodology section. The authors need to make clarifications for the below comments:
In Line 123, it is not clear why the authors have used Saifur et al. (2022), is this a reference if so, then it’s not included in the reference section; if not, then the sentence has to be revised.
In Line 125, the authors mentioned, 'they provided one ml of an allantoic fluid from an infected egg.' It’s not clear who has provided the allantoic fluid.

In the result section
Line 208, change Eg20616 to Eg2061.

Validity of the findings

The study design , results & conclusion of this manuscript are well defined with supporting results.

Additional comments

no comments

---

## Round 0.2 · accepted · Accept

Authors have addressed the reviewer comments. I am satisfied with the revised manuscript.

Reviewer 2 ·

Basic reporting

Clear and well-defined. The authors have addressed all the queries raised.

Experimental design

The authors have addressed and modified the methodology section as requested

Validity of the findings

The conclusion has been well stated